# Transcriptional dynamics in the protozoan parasite *Sarcocystis neurona* and mammalian host cells after treatment with a specific inhibitor of apicomplexan mRNA polyadenylation

**Arthur G. Hunt** [ID] [1] *, **Daniel K. Howe** [ID] [2], **Ashley Brown** [2], **Michelle Yeargan** [2]

1 Department of Plant and Soil Sciences, University of Kentucky, Lexington, KY, United States of America,
2 Department of Veterinary Science, University of Kentucky, Lexington, KY, United States of America

* aghunt00@uky.edu

**Data Availability Statement:** The high throughput sequencing data associated with this project are

## Abstract

In recent years, a class of chemical compounds (benzoxaboroles) that are active against a range of parasites has been shown to target mRNA polyadenylation by inhibiting the activity of CPSF73, the endonucleolytic core of the eukaryotic polyadenylation complex. One particular compound, termed AN3661, is active against several apicomplexan parasites that cause disease in humans. In this study, we report that AN3661 is active against an apicomplexan that causes disease in horses and marine mammals (*Sarcocystis neurona*), with an approximate $IC_{50}$ value of 14.99 nM. Consistent with the reported mode of action of AN3661 against other apicomplexans, *S. neurona* mutants resistant to AN3661 had an alteration in CPSF73 that was identical to a mutation previously documented in AN3661-resistant *Toxoplasma gondii* and *Plasmodium falciparum*. AN3661 had a wide-ranging effect on poly(A) site choice in *S. neurona*, with more than half of all expressed genes showing some alteration in mRNA 3' ends. This was accompanied by changes in the relative expression of more than 25% of *S. neurona* genes and an overall 5-fold reduction of *S. neurona* transcripts in infected cells. In contrast, AN3661 had no discernible effect on poly(A) site choice or gene expression in the host cells. These transcriptomic studies indicate that AN3661 is exceedingly specific for the parasite CPSF73 protein, and has the potential to augment other therapies for the control of apicomplexan parasites in domestic animals.

## Introduction

The phylum Apicomplexa encompasses a very broad group of obligate intracellular parasites that are a significant cause of disease worldwide [1]. Several of these are important human pathogens, such as *Plasmodium* spp. and *Toxoplasma gondii*. As well, multiple members of this phylum cause disease in domestic animals and have a significant economic impact on agriculture. Both *Theileria* spp. and *Babesia* spp. are tick-borne haemoprotozoan pathogens that

available under Bioproject PRJNA713353. https://
www.ncbi.nlm.nih.gov/bioproject/PRJNA713353.

**Funding:** This research was supported by the
USDA National Institute of Food and Agriculture
(AGH - Hatch project accession #1020849 and
DKH – Hatch project accession 1012262; https://
nifa.usda.gov/) and the Amerman Family Equine
Research Fund (DKH). The sponsors played no
role in the study design, data collection and
analysis, decision to publish, or preparation of the
manuscript.

**Competing interests:** The authors declare no
competing interests.

infect and cause disease in cattle and horses. Species of *Eimeria* are enteric parasites that cause
diarrheal diseases in a variety of animals and are especially detrimental to the poultry industry.
*Neospora caninum* was initially described as a neurologic pathogen of canids, but has subse-
quently been identified as a major cause of reproductive failure in cattle. *Sarcocystis neurona* is
an important cause of the neurologic disease equine protozoal myeloencephalitis (EPM) and
an emerging pathogen of marine mammals. While the economic impact of these pathogens is
difficult to assess, the annual global cost of neosporosis alone has been estimated at greater
than $1 billion dollars [2].

Although vaccination is an option for reducing coccidiosis caused by *Eimeria* spp. in poul-
try, vaccines against other coccidian parasites have generally proven ineffective. Consequently,
chemotherapy has been the standard approach to try to control infection and disease caused
by these parasites (reviewed in [3–5]). Drug combinations that target folate metabolism, such
as pyrimethamine and a sulfonamide, have been used for decades to treat infections by multi-
ple apicomplexan parasites, including the coccidia. However, the routine use of anti-folate
drugs to treat coccidial infections has been hindered by several factors, including the duration
of treatment (6 months for horses with EPM) and associated toxicities (anemia, leukopenia,
teratogenicity). The benzeneacetonitrile compounds diclazuril, ponazuril, and toltrazuril are
effective treatments for agricultural animals infected with coccidian parasites. However, con-
cerns about drug residues in food products and the ease in developing resistance have limited
the use of the benzeneacetonitrile drugs in agricultural animals [3–5]. Anticoccidial medica-
tions are used extensively against *Eimeria* and are quite commonly incorporated into poultry
feeds. A variety of drugs have been employed for this purpose, but ionophore antibiotics, such
as monensin, are currently the most widely used anticoccidials to control coccidiosis (reviewed
in [6]). While resistance to the ionophore drugs has been slower to occur, reports now indicate
that monensin-resistant isolates of *Eimeria* are widespread [6]. Consequently, the continued
use of existing anticoccidial drugs to control these parasites might not be a viable long-term
option. Taken together, these considerations reveal an ongoing need for affective alternative
therapies for coccidian diseases in agricultural animals.

Recently, it was reported that a member of a class of benzoxaboroles, termed AN3661,
inhibits growth of *Plasmodium falciparum* [7], *T. gondii* [8], and *Cryptosporidium parvum* [9].
The demonstrated mode of action of AN3661 is novel, in that it targets the 73 kDa subunit of
the Cleavage and Polyadenylation Specificity Factor, or CPSF73, of the apicomplexan polyade-
nylation complex. In this report, we show that AN3661 similarly inhibited the coccidian para-
site *S. neurona*. We also show that AN3661 had a decided effect on transcriptional dynamics of
*S. neurona*, while drug treatment of the mammalian host cells had little impact on gene expres-
sion and mRNA polyadenylation in these cells. Together, these results expand the list of api-
complexans species that are susceptible to AN3661, and they reveal that the drug is an
exceedingly selective inhibitor, with little discernible impact on the gene expression dynamics
of the mammalian host.

## Materials and methods

### Parasite cultures, DNA isolation, and parasite growth assays

*S. neurona* strains (wild-type strain SN3.E1, YFP-expressing clone F9F, and AN3661-resistant
clones were propagated by serial passage in monolayers of BT cells, as described [10]. Upon
lysis of the infected monolayers, extracellular merozoites were harvested by passing through 23
and 25-gauge needles and filter-purified to remove host cell debris. Merozoites were pelleted
and stored at -80°C until used for DNA or RNA isolation.

For parasite growth assays, YFP-expressing *S. neurona* was used according to procedures described previously [11]. Freshly-released merozoites purified from host cell debris were resuspended in culture medium without phenol red, and 96-well plates containing BT mono-layers were inoculated with $4 \times 10^4$ parasites per well, eight replicates (wells) per treatment. Control wells containing no parasites or no drug were included to account for background fluorescence and relative growth, respectively. Non-invaded merozoites were washed out after 2 hrs, and medium containing the appropriate dilution of AN3661 was added back to the wells and left for the duration of the growth assay. The plates were incubated at 37°C for 4 days, and fluorescence was measured using a Synergy H1 plate reader (BioTek, Winooski, VT, USA). The relative fluorescence units (RFUs) in the no-parasite wells was subtracted from the no-drug control and treatment wells, and growth of *S. neurona* in the presence of AN3661 was determined by comparing the RFUs in treated wells with those from non-treated control wells. The half-maximal inhibitory concentration ($IC_{50}$) was determined by regression analysis (GraphPad Prism v. 9).

## Isolation and sequence analysis of AN3661-resistant clones of *S. neurona*

*Sarcocystis neurona* strain SN3.E1 was mutagenized with 2 mM N-ethyl-N-nitrosourea (ENU), as described previously for *T. gondii* [12]. The mutagenized culture was allowed to recover and expand for 3 days before addition of 90 nM AN3661 to select for mutant parasites that had become resistant to the drug. The *S. neurona* culture was maintained in medium containing AN3661 until the parasites disrupted the host cell monolayer (approximately 5 weeks), and single-cell clones of AN3661-resistant parasites were isolated in 96-well plates and expanded in the presence of drug for further analyses, as described [13].

The full-length coding region of SnCPSF73 was amplified in sections and sequenced using the primers listed in S1 Table. Sequences obtained from the AN3661-resistant clones were aligned with the SnCPSF73 gene from the SN3.E1 reference genome (SN3_01500330) to iden-tify nucleotide polymorphisms. The SnCPSF73 amino acid sequence was further aligned to the human (NP_057291.1), *Arabidopsis* (At1g61010.1), *T. gondii* (TGME49_285200-t26_1), and *P. falciparum* (PF3D7_1438500.1) CPSF73 amino acid sequences so as to display commonalities in mutations that arise in AN3661-resistant *S. neurona* clones.

## Poly(A) site profiling

*S. neurona* strain SN3.E1 was inoculated onto BT host cell monolayers grown in 6-well plates. After 48 hrs of parasite development, triplicate wells (each well representing a biological repli-cate) were incubated in fresh media with or without 90 nM AN3661 for an additional 24 hrs. The infected BT monolayers were then harvested from the wells using cell scrapers and centri-fuged at 1100 x g for 10 min at 4°C. The cell pellets were washed 1x with ice-cold PBS and then stored at -80°C until used for RNA isolation. To conduct poly(A) site profiling in the non-infected host cells, confluent monolayers of BT cells grown in 6-well plates, triplicate wells were similarly treated with or without AN3661 for 24 hrs, and the cells harvested for RNA isolation.

Total RNA was isolated by adding 1mL Trizol to cell pellets, incubating for 5 minutes at room temperature, then adding 200 μL chloroform and incubating for an additional 3 minutes at room temperature. Samples were centrifuged at 13,200 x g for 15 minutes, and the aqueous layer was transferred to a fresh tube. 500 μL of isopropanol was added to the sample and incu-bated overnight at -20°C. The tubes were centrifuged at 13,200 x g for 10 minutes, and the RNA pellet was washed with 1 mL 75% EtOH. The pellet was air dried for 5–10 minutes and resuspended in RNase free water.

Libraries for genome-wide poly(A) site profiling was performed following the procedures described in Ma *et al.* [14] and Pati *et al.* [15]. So-called poly(A) tag (PAT-Seq) libraries were sequenced on an Illumina HiSeq instrument; the sequencing data are available under Bioproject PRJNA713353. Sequencing reads were trimmed, demultiplexed, and mapped to the respective genomes using CLC Genomics Workbench (latest version used was 20.0.4); the results of these analyses are summarized in S1 File. Subsequent analyses were performed following the pipelines described in Bell *et al.* [16] and de Lorenzo *et al.* [17]. This pipeline is described in detail in S2 File.

Gene expression was estimated by mapping individual PAT-Seq reads to the *S. neurona* or *B. taurus* genome annotations using the RNASeq tool in CLC Genomics Workbench. The process is described in detail in S2 File.

### Host cell cytotoxicity assay

The cytotoxicity of AN3661 for the bovine host cells was determined by release of lactate dehydrogenase (LDH) from treated cells using a colorimetric assay (Pierce LDH Cytotoxicity Assay, Thermo Scientific). Briefly, BT cells seeded in a 96-well plate were treated in triplicate in increasing concentrations of AN3661. The plate was incubated for 24 hours, and LDH release from the treated cells was determined spectrophotometrically by subtracting the 680 nm absorbance (background) from the 490 nm absorbance following the protocol provided by the manufacturer. The maximum release of LDH was determined by addition of lysis buffer (provided by the manufacturer) to non-treated triplicate wells of BT cells. The LDH positive control was provided by the manufacturer.

## Results

### AN3661 inhibits the growth of *Sarcocystis neurona* and *Neospora caninum*

To test the hypothesis that AN3661 inhibits *S. neurona*, growth assays were performed using a parasite clone expressing YFP, as described previously [11,18]. Our preliminary growth assays suggested that AN3661 inhibited *S. neurona* at low nanomolar concentrations (data not shown). Based on these initial assays, *S. neurona* growth was examined in the presence of AN3661 over a range of 1 nM to 100 nM (Fig 1A). This showed that *S. neurona* is very sensitive to AN3661, with an estimated $IC_{50}$ of 10.68 nM and nearly all parasite growth inhibited at concentrations greater than 50 nM (Fig 1A). To obtain a better estimate of the AN3661 $IC_{50}$ for *S. neurona*, the growth assay was repeated using a narrower range of drug concentration (1–25 nM). This assay revealed an $IC_{50}$ of 14.99 nm (Fig 1B), a concentration comparable to the $IC_{50}$ described for two isolates of *P. falciparum* [7], but significantly less than the $IC_{50}$ reported for *C. parvum* (80 nM [9]) and *T. gondii* (900 nM [8]).

### AN3661-resistant *S. neurona* have mutations near the active site of CPSF73

To further explore the mechanism of inhibition of *S. neurona* by AN3661, parasites were treated with the chemical mutagen N-ethyl-N-nitrosourea (ENU) and grown in the presence of 90 nM AN3661, a concentration of the drug that was found to be highly effective for inhibiting growth of the parasite (Fig 1A). Seven drug-resistant single-cell clones were isolated, and five were successfully used for further analyses. Based on the prior studies indicating that CPSF73 is the target of AN3661, the coding region of this gene (SN3_01500330) was amplified by PCR, sequenced, and the sequences aligned with the wild-type *S. neurona* CPSF73 coding sequence. These comparisons revealed that all five clones carried a point mutation resulting in a Y668N change in the *S. neurona* protein (Fig 2), a position that is also altered in

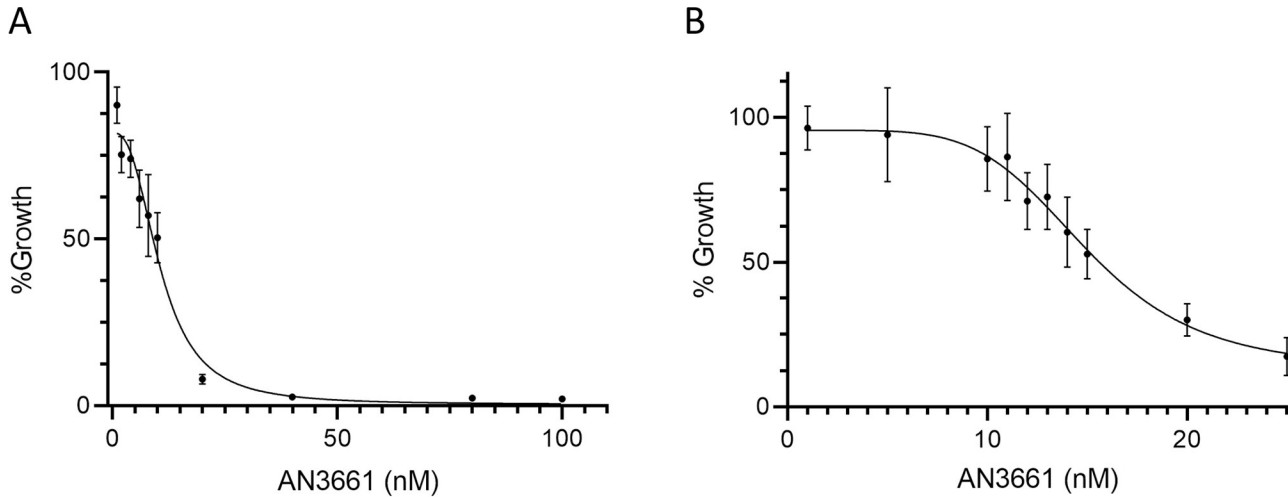

**Fig 1. Growth inhibition of *S. neurona* by AN3661.** An *S. neurona* clone expressing YFP was seeded into 96-well plates containing monolayers of BT host cells and incubated for 4 days (*S. neurona*) in the presence of increasing concentrations of AN3661 (eight replicates/concentration). Parasite growth inhibition was based on relative fluorescence of treated wells compared to control wells containing no drug, and the IC$_{50}$ was determined by regression analysis. (A) Parasite growth in AN3661 concentrations between 1 nM and 100 nM suggested an IC$_{50}$ of 10.68 nM, with parasite growth virtually halted at concentrations greater than 50 nM. (B) Parasite growth across a narrow range of AN3661 concentrations indicated an IC$_{50}$ of 14.99 nM.

AN3661-resistant *T. gondii* and *P. falciparum* [7,8]. While it is possible that each clone may possess additional mutations, the consistent alteration of Y668 in five independent clones nonetheless supports the hypothesis that AN3661 targets CPSF73 in *S. neurona*, much as it does in other apicomplexans.

## Effects of AN3661 on gene expression and poly(A) site choice in *S. neurona*

To assess the effects of AN3661 on transcription dynamics of *S. neurona*, genome-wide transcription was assessed by preparing and sequencing 3' end-directed cDNA tags (so-called PATSeq), and subsequent analyses. For this, bovine turbinate (BT) cell cultures were infected with *S. neurona* in triplicate and grown with or without 90 nM AN3661, a concentration of drug found to fully inhibit parasite growth (Fig 1A). Between 3 and 6 million reads were returned for each library (S1 File). After demultiplexing and trimming, the reads were mapped to the *S. neurona* SN3.E1 (GenBank accession GCA_000727475.1) and *Bos taurus* (Hereford) genomes (ARS-UCD1.2); the latter mapping was used to assess the effects of the inhibitor on transcriptional dynamics in the host cells (following section). To confirm the reproducibility of the libraries, global gene expression measurements were conducted and gene-by-gene comparisons or relative expression levels were made and presented as the Pearson correlation coefficients for each pairwise comparison. Values for these coefficients ranged from 0.96 to 0.99, with one exception (which yielded a coefficient of 0.85; S1 File).

To assess the overall effects of the CPSF73 inhibitor on transcriptional output, the relative quantities of reads that map to the two genomes were compared in control and AN3661-treated samples. As shown in Fig 3A, more than 60% of the reads mapped to the *S. neurona* genome in the non-treated cultures. In contrast, in AN3661-treated cultures, only about 25% of reads mapped to the parasite genome. This result indicates an almost 5-fold reduction in the overall proportion of transcripts from *S. neurona* that had been treated with the inhibitor, and is consistent with the growth characteristics of the parasite under these conditions (Fig 1A).

```
Tg CPSF73  LGACMFLIEI  GGVRMLYTGD  FSRESDRHVP  IAEVPPVDVQ  LLICESTYGI  HVHDDRQLRE  340
Sn CPSF73  LGACMFLVET  GGVRTLYTGD  FSSEEDRHVP  AAELPAVDVQ  LLLCESTYGV  QVHEPRQQRE  522
Pf CPSF73  IGACMFLVEI  NNIRFLYTGD  YSREIDRHIP  IAEIPNIDVH  VLICEGTYGI  KVHDDRKKRE  264
At CPSF73  LGAAMFMVDI  AGVRILYTGD  YSREEDRHLR  AAELPQFSPD  ICIIESTSGV  QLHQSRHIRE  229
Hs CPSF73  LGAAMFMIEI  AGVKLLYTGD  FSRQEDRHLM  AAEIPNIKPD  ILIIESTYGT  HIHEKREERE  219

Tg CPSF73  RRFLKAVVDI  V-NRGGKCLL  PVFALGRAQE  LLLILEEYWT  AHPEIRHVPI  LFLSPLSSKC  399
Sn CPSF73  SRLIAAILEI  VLQRRGKVLL  PVFALGRVQE  LLLILEEYWR  SHPAVQHVPI  LFISPLASKS  582
Pf CPSF73  IRFLNILTSM  I-NNKGKVLL  PVFALGRAQE  LLLILEEHWD  KNKHLQNIPI  FYISSMATKS  323
At CPSF73  KRFTDVIHST  V-AQGGRVLI  PAFALGRAQE  LLLILDEYWA  NHPDLHNIPI  YYASPLAKKC  288
Hs CPSF73  ARFCNTVHDI  V-NRGGRGLI  PVFALGRAQE  LLLILDEYWQ  NHPELHDIPI  YYASSLAKKC  278

Tg CPSF73  AVVFDAFVDM  CGEAVRSRAL  RGENPFAFRF  VKNVKSVE--  AARVYIHHDG  PAVVMAAPGM  457
Sn CPSF73  IAVFDAFLHM  SGRQLRQQAL  QGENPFAFQF  VKSLKSYDGA  AARLYVHRDA  PAVIFAAPGM  642
Pf CPSF73  LCIYETFINL  CGEFVKKVVN  EGKNPFNFKY  VKYAKSLESI  SSYLY-QDNN  PCVIMASPGM  382
At CPSF73  MAVYQTYILS  MNDRIRNQ-F  ANSNPFVFKH  ISPLNSID--  ----DFNDVG  PSVVMATPGG  341
Hs CPSF73  MAVYQTYVNA  MNDKIRKQ-I  NINNPFVFKH  ISNLKSMD--  ----HFDDIG  PSVVMASPGM  331

Tg CPSF73  LQSGASREIF  EAWAPDAKNG  VILTGYSVKG  TLADELKREP  ETIQLP----  ----------  503
Sn CPSF73  LQSGASRRIF  EVLAPHARNG  VILTGYSVKG  TLAEELKREP  EHLLLPRSDA  GGGGGAGAAA  702
Pf CPSF73  LQNGISKNIF  NIIASDKKSG  VILTGYTVKG  TLADELKTEP  EFVTI-----  ----------  427
At CPSF73  LQSGLSRQLF  DSWCSDKKNA  CIIPGYMVEG  TLAKTIINEP  KEVTL-----  ----------  386
Hs CPSF73  MQSGLSRELF  ESWCTDKRNG  VIIAGYCVEG  TLAKHIMSEP  EEITT-----  ----------  376

Tg CPSF73  ----------  -----DRVLR  RRCSFEMISF  SAHSDYQQTQ  EFIGKLKVPN  VVLVHGERGE  548
Sn CPSF73  AGGAATGAET  DLQPQQQVLR  RRCSCQVISF  SAHSDYLQTS  AFIKQLRVPN  VVLVHGERNE  762
Pf CPSF73  ----------  ----NDKVVK  RKCRFEQISF  SAHSDFNQTK  TFIEKLKCPN  VVLVHGDKNE  473
At CPSF73  ----------  ---MNGLTAP  LNMQVHYISF  SAHADYAQTS  TFLKELMPPN  IILVHGEANE  433
Hs CPSF73  ----------  ---MSGQKLP  LKMSVDYISF  SAHTDYQQTS  EFIRALKPPH  VILVHGEQNE  423
```

**Fig 2. Amino acid sequence alignments of orthologs of CPSF73.** Alignments were performed using the default settings in CLC Genomics Workbench. Amino acid residues that are identical in all five sequences are shown in black, and other residues in gray. Residues that are altered in *T. gondii* AN3661-resistant mutants are denoted with a blue arrow, those altered in resistant *P. falciparum* mutants in red, and those altered in both *T. gondii* and *P. falciparum* mutants in green. The green star denotes the position (Y668 in the *S. neurona* protein, corresponding to Y483 of the *T. gondii* CPSF73) that is altered in the five AN3661-resistance *S. neurona* clones. The alignment is truncated, focusing on the conserved core of the protein. Tg–*T. gondii*; Sn–*S. neurona*; Pf–*P. falciparum*; At–*Arabidopsis thaliana;* Hs–human. Numbers on the right denote the residue numbers for the respective proteins.

This analysis was augmented by a genome-wide analysis of parasite gene expression in control and inhibitor-treated cultures. As shown in Fig 3B and S3 File, in inhibitor-treated cells, 1253 genes showed a minimum of a 2-fold change in gene expression with false discovery rate adjusted p-values of 0.05 or less. This represents 18% of all annotated genes in the *S. neurona* SN3.E1 genome version used for this analysis, and 22% of all genes that showed some degree of gene expression. Genes encoding enzymes and proteins associated with protein biosynthesis were over-represented in the set of differentially-expressed genes (S3 File). This probably reflects a general effect of the inhibitor on transcriptional output and protein synthesis, and concomitant adjustments by the parasite.

Given the target of AN3661 (namely, CPSF73), these global transcriptomics analyses were supplemented with a study of the drug's effects on poly(A) site choice in the parasite. For this, the analysis pipeline described earlier [16,17,19] was adapted to identify individual poly(A) sites whose relative usage was affected by the inhibitor. In this analysis, relative usage is defined as the fraction of all reads mapping to a given gene that also mapped to a given poly(A) site. This approach is biased towards identifying changes in sites whose usages may be low, and does not apply to genes that have only one poly(A) site. Nonetheless, it serves as a useful metric

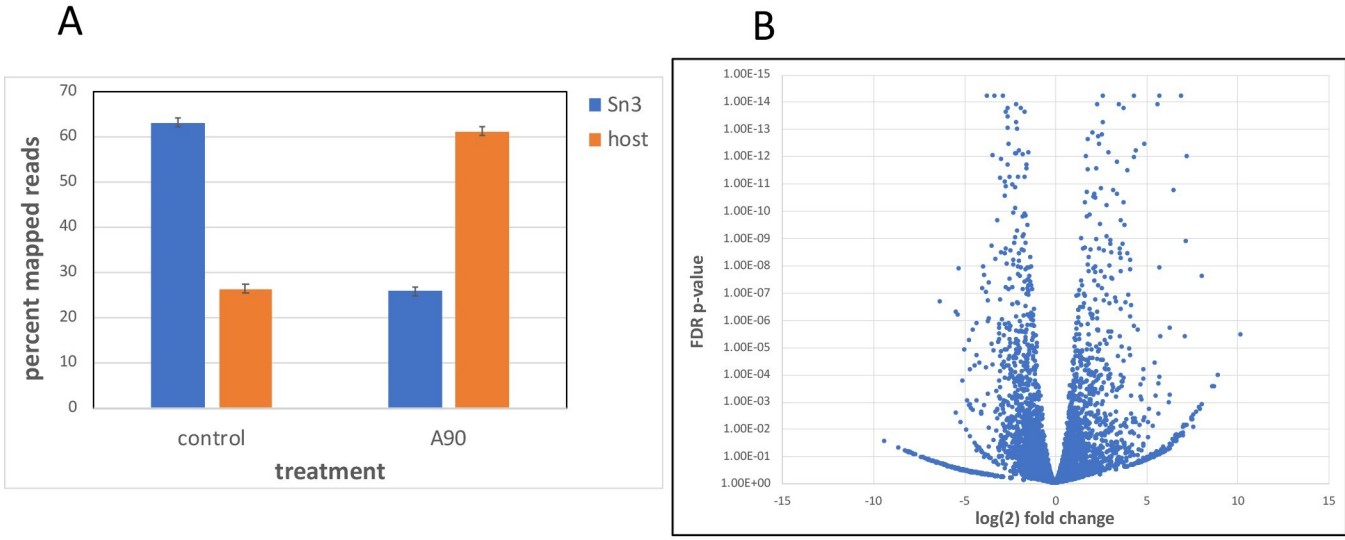

**Fig 3. Gene expression in control and AN3661-treated cells.** A. Overall parasite and host expression in control and AN3661-treated cells. The percent of all reads that map to the *S. neurona* and *B. taurus* genomes was calculated and plotted as shown. "control"–infected cells grown in the absence of AN3661. "A90"– infected cells treated with 90 nM AN3661. Reads mapping to the parasite genome are summarized with blue bars, and those mapping to the host cell genome with orange bars. Error bars denote standard deviations calculated from the values in each replicate of the respective sample. B. Volcano plot depicting changes in *S. neurona* gene expression due to AN3661 treatment.

to assess effects on poly(A) site choice, the rationale being that changes that affect the core machinery will result in increased usage of minor, non-canonical poly(A) sites. The results of this analysis (S4 File) revealed that 1363 individual sites in 990 genes showed significant changes in poly(A) site usage using an false discovery rate (FDR)-adjusted p-value cutoff of 0.05. A more lenient cutoff (raw p-value < 0.05) yielded 5730 poly(A) sites in 2763 genes whose usage was altered in AN3661-treated cells. This is a substantial fraction of all genes (5955) with detectable levels of expression, and indicates a wide-ranging impact of the inhibitor on mRNA 3' end formation.

To assess the possible contribution of this altered poly(A) site choice to overall gene expression levels, the sets of genes whose expression change were compared with those affected by changes in poly(A) site choice. The results (Fig 4) show that between 24 and 27% of genes whose poly(A) site profiles change also showed significant changes in overall transcript levels. These results suggest that alternative polyadenylation (APA) may contribute to the overall pattern of gene expression in inhibitor-treated cells.

An inhibitor of mRNA polyadenylation is expected to alter transcription termination and increase the production of readthrough transcripts that might be "rescued" (and manifest as polyadenylated transcripts) through the use of cryptic or non-canonical polyadenylation sites. The latter would be apparent as an increased number of PATSeq reads mapping to unannotated parts of the genome or that are antisense in orientation to annotated genes. When these reads were measured, it was apparent that AN3661 increased by almost two-fold the overall levels of PATSeq reads that mapped outside of annotated regions (Fig 5A) or antisense to annotated genes (Fig 5B). The summed impact was that almost 10% of the total transcriptional output appeared to be associated with non-canonical poly(A) sites (Fig 5C). (Examples that illustrate these changes are shown in S1 Fig). This result corroborates those shown in S4 File and together support the hypothesis that AN3661 has a sizeable impact on transcription by altering mRNA polyadenylation.

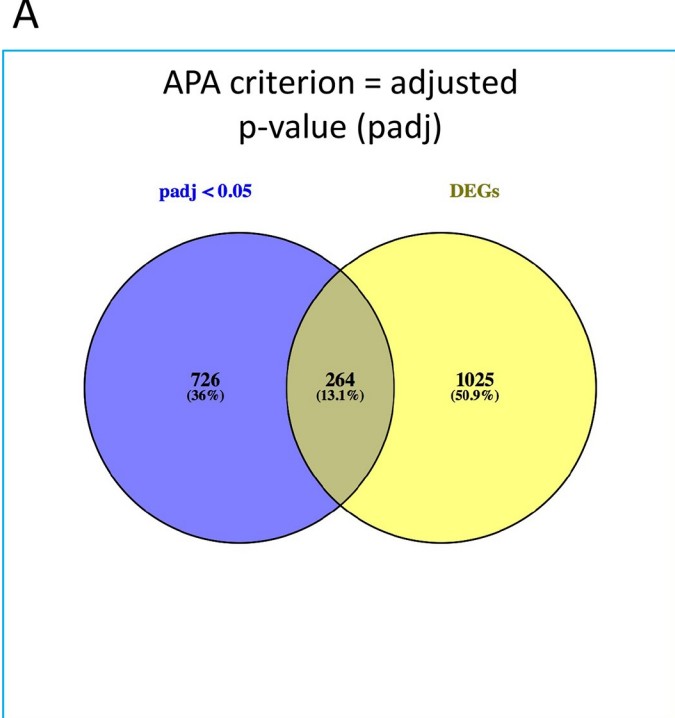
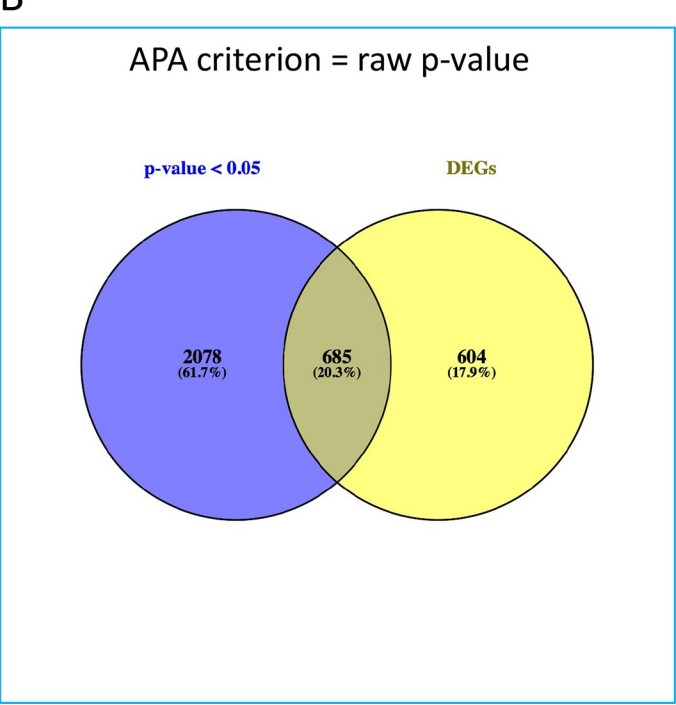

**Fig 4. Genes that show differential poly(A) site usage and overall expression.** Venn diagrams showing the overlaps between genes affected by APA (using the padj or p-value cutoffs described in the text, from S4 File; blue circles) and differentially expressed genes (from S3 File; yellow circles). Venn diagrams were made using Venny [20].

## Effects of AN3661 on mRNA polyadenylation and gene expression in the BT host cells

In other systems, AN3661 has a high degree of specificity, with strong inhibition of the growth of *P. falciparum*, *T. gondii*, and *C. parvum* but little discernible effects on their respective hosts. To confirm that this is the case with the host cells used here for growth of *S. neurona*,

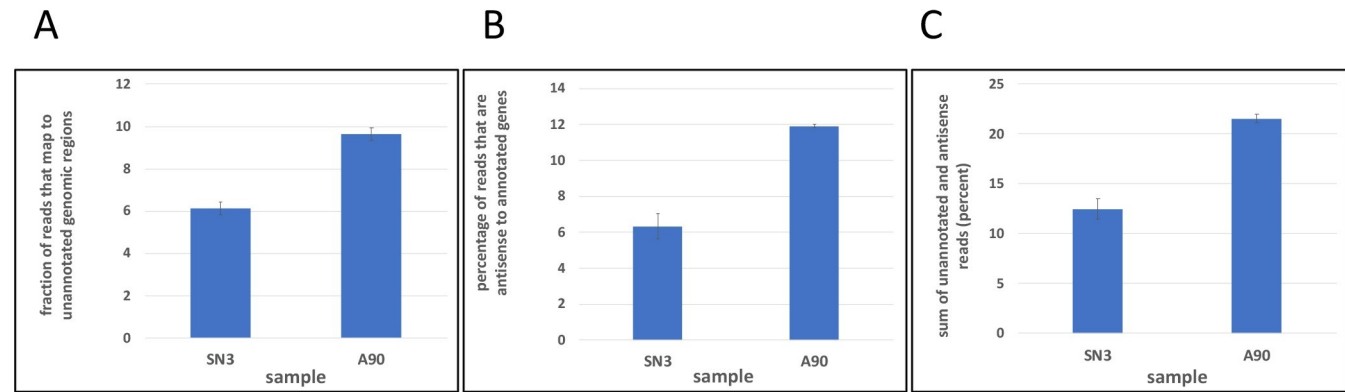

**Fig 5. Fraction of all mapped reads that map outside of or antisense to annotated genes.** Fraction (represented as percentages) of all mapped reads that map outside of or antisense to annotated genes. SN3 –untreated control cells. A90 –cells treated with 90 nM AN3661. Error bars denote standard deviations calculated from the values in each replicate of the respective sample. A. Reads that map outside of annotated regions. B. Reads that map antisense to annotated genes. C. Sum of data shown in panels A and B.

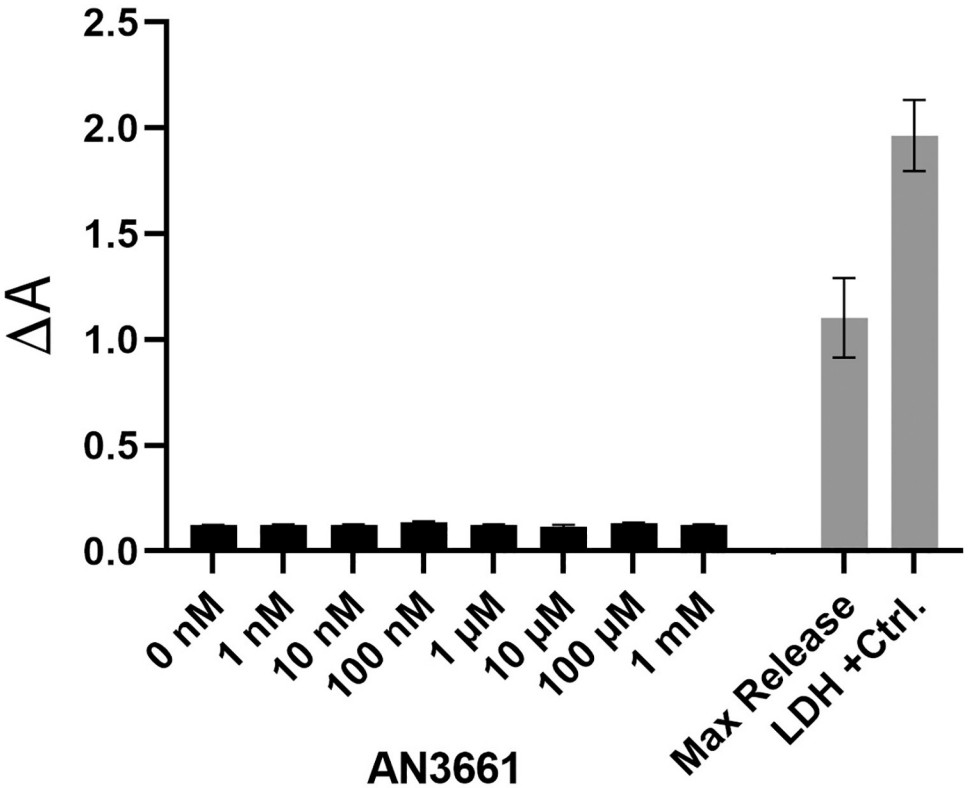

**Fig 6. Cytotoxicity of AN3661 for bovine turbinate BT cells determined by LDH release.** Triplicate wells of BT cultures in a 96-well plate were treated for 24 hours with drug concentrations from 1 nM to 1 mM, and the level of LDH released into the medium was compared to the LDH positive control provided by the manufacturer (Thermo Scientific), the maximum LDH release (i.e., lysed cells), and the spontaneous LDH release (non-treated cells). The LDH activity present in the samples was determined spectrophotometrically by subtracting the absorbance at 680 nm (background) from absorbance at 490 nm. Thus, $\Delta A = A490 \text{ nm} - A_{680 \text{ nm}}$, where A is absorbance.

cytotoxicity and gene expression was assessed in BT cells that were treated with AN3661. In a standard LDH release assay that measures cytotoxicity (Pierce LDH Cytotoxicity Assay), the LDH levels in media prepared from BT cells treated with AN3661 concentrations ranging from 1 nM to 1 mM were indistinguishable from LDH released from non-treated cells (0 nM), and were only slightly more than 10% of the maximum release (lysed cells) control (Fig 6). Therefore, AN3661 was not cytotoxic to the BT host cell line, even at concentrations approaching 100,000-fold greater than the IC$_{50}$ for *S. neurona* (14.99 nM; Fig 1B).

To assess the effects of AN3661 on gene expression in the BT host cells, PATSeq libraries were prepared from uninfected cells grown in the absence or presence of 90 nM AN3661 (the same concentration as was used to study gene expression in *S. neurona*-infected cells). These libraries were sequenced using the Illumina platform and mapped to the *B. taurus* (Hereford) genome. Consistency between replicates was assessed by comparisons of genome-wide expression in individual replicates. These pairwise comparisons of libraries prepared from uninfected host cells yielded consistently high Pearson Correlation coefficients (uniformly greater than 0.96; S1 File). Given this, several additional analyses were conducted to assess the impacts of AN3661 on gene expression and polyadenylation.

As was done for the analysis of *S. neurona* gene expression, gene-by-gene expression in treated and control host cells was measured. The results showed that only 78 genes (of 14,785 whose expression could be measured in this study) had significantly-different expression

A

B

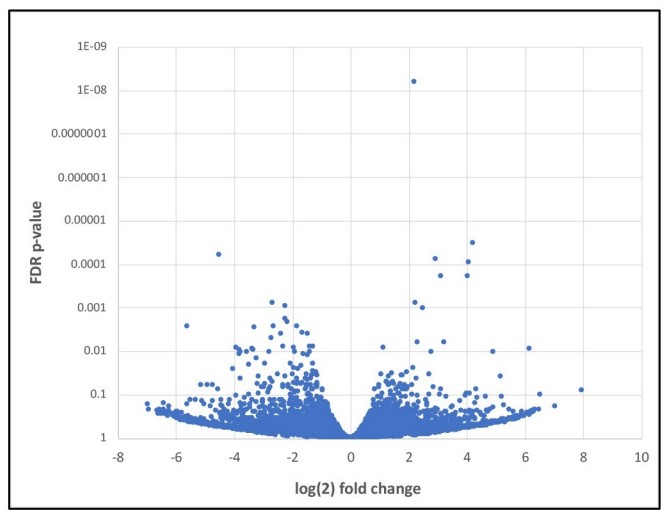

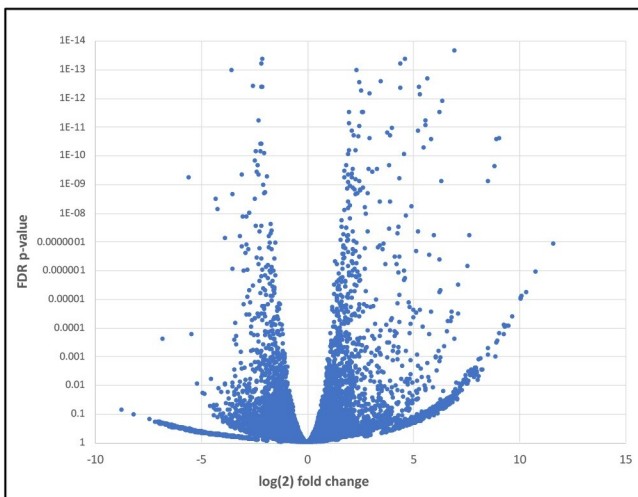

**Fig 7. Volcano plots showing the extent of differential host gene expression in infected and AN3661-treated cells.** A. Comparison of host cell gene expression in untreated and AN3661-treated cells. B. Comparison of host cell gene expression in uninfected and *S. neurona*-infected cells.

(2-fold or greater change in expression, with an FDR-adjusted p-value less than 0.05) in AN3661-treated BT host cells (Fig 7A, S5 File). By way of comparison, *S. neurona* incited changes in the expression of 1489 genes in the host cells (Fig 7B, S6 File). Together, these results indicate that AN3661 has a minimal impact on overall gene expression in BT host cells.

In parallel, the effect of AN3661 on poly(A) site choice in uninfected BT cells was assessed using the analysis pipeline described in S2 File. A site-by-site analysis of poly(A) site usage in control and treated cells showed that none of the >35,000 sites for which results could be returned showed changes that satisfied the FDR-adjusted p-value cut-off of 0.05 (S7 File). Using the more lenient raw p-value cutoff of 0.05 (as was done for the analysis of *S. neurona* poly(A) site choice described above), 32 sites from 31 genes were returned. These results indicate minimal effect of AN3661 on mRNA polyadenylation in BT cells.

To further explore possible effects on transcription, the numbers of PATSeq reads that map outside of, or antisense to, annotated BT genes was tabulated. The result of this exercise showed that there were no appreciable differences in the quantities of such reads between control and treated BT cells (Fig 8). Together, these results indicate that AN3661 has almost no impact on gene expression, mRNA polyadenylation, and transcriptional readthrough in BT cells.

## Discussion

### AN3661 targets CPSF73 and mRNA polyadenylation in *S. neurona*

In recent years, a variety of benzoxaboroles have been found to target mRNA processing in a number of unicellular parasites, including several that can cause disease in humans [7–9,21–23]. The benzoxaborole AN3661, the subject of this study, inhibits *P. falciparum*, *T. gondii*, and *C. parvum* by targeting CPSF73, a subunit of the parasite polyadenylation complex [7–9]. The results presented in this study extend the range of apicomplexan parasites that are inhibited by AN3661 to include two species that have importance in agriculture. Moreover, the

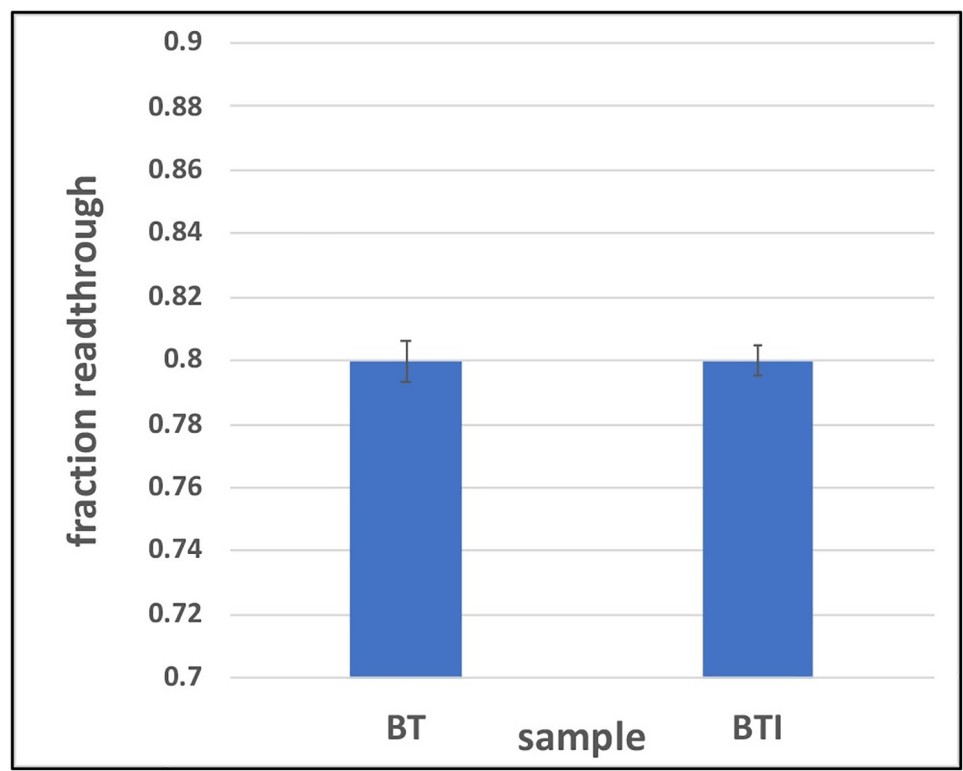

**Fig 8. Transcriptional readthrough in BT host cells.** Fraction of all mapped reads that map outside of or antisense to annotated genes in control and AN3661-treated host cells was measured and plotted as shown. BT–untreated control cells. BTI–cells treated with 90 nM AN3661. For this plot, the fractions of reads that map outside of and antisense to annotated genes were summed and plotted as shown. Note the scale of the y-axis.

finding that AN3661-resistant *S. neurona* clones carry mutations in the gene that encodes CPSF73 corroborates earlier studies indicating that this drug targets the mRNA polyadenylation in apicomplexan parasites.

While the target of AN3661 and the physiological consequences of treatment with the drug have been well-established [7–9], a detailed understanding of the consequences of AN3661 treatment on genome-wide transcription dynamics in apicomplexans parasites is lacking. The results presented in this report provide added insight into the effects of the drug on 3' end processing and transcription termination. Treatment with the drug results in a general diminution of parasite transcript levels in infected cells (Fig 3A). This is consistent with the growth inhibition attendant with drug treatment (Fig 1). Interestingly, some genes are more affected by drug treatment than others, resulting in a wide-ranging change in relative gene expression in the two conditions (Fig 3B). These changes in all likelihood reflect the differential responses of gene expression to the physiological changes brought about by the alterations in mRNA polyadenylation incited by the drug.

Along with the overall transcript-level response, treatment with AN3661 also leads to a global alteration in poly(A) site choice in *S. neurona*, affecting almost half of all expressed genes in the parasite. This is very likely a consequence of a general inhibition of 3' end processing by the drug. AN3661 binds in the active site of the apicomplexan CPSF73 in a manner predicted to interfere with the metal ion-mediated catalytic mechanism of the endonuclease [9]. Inhibition of processing at the primary (preferred) poly(A) site is expected to result in a population of extended primary transcripts, the collection of which would present numerous

alternate poly(A) signals to the processing apparatus. Any of these sites might be recognized and processed, guided by the dynamics of the interactions between the nascent RNA and the poly(A) complex. Weak sites that are usually skipped might be utilized at greater frequencies in drug-treated cells. There is increased read-through transcription in drug-treated *S. neurona* cells (Fig 5), and additional potential sites might be present in extended transcripts. Taken together, the global shifts in poly(A) site choice upon AN3661 treatment are consistent with the reported mode of action of the drug [7–9].

In these regards, the effects of AN3661 on polyadenylation in *S. neurona* are similar to the consequences of reduction of CPSF73 levels or activity in mammalian cells and in yeast. The mammalian CPSF73 is a target of a compound (JTE-607) that is active against acute myeloid leukemia and Ewings sarcoma cell lines, and acts by mimicking RNA binding at the CPSF73 active site [24,25]. In JTE-607 treated cells, a substantial increase in read-through transcription is seen [25], much as is seen in AN3661-treated *S. neurona* cells (Fig 5). This parallel is supportive of the model for the proposed mechanism of action of AN3661 in *S. neurona*, namely that the compound inhibits CPSF73 and alters efficient mRNA polyadenylation and transcription termination.

YSH1 is the yeast counterpart of CPSF73 and resides in a complex that includes other CPSF subunit orthologs [26]. Like the mammalian CPSF73, YSH1 is the endonuclease that cleaves the pre-mRNA prior to polyadenylation. YSH1 levels are maintained in part by the action of the *IPA1* gene product, such that *ipa1* mutants have substantially-reduced levels of YSH1 [27]. *ipa1* mutants exhibit substantial degrees of transcriptional readthrough, and also considerable changes in the usage of annotated poly(A) sites [28]. These are features also seen in AN3661-treated *S. neurona* cells. This parallel provides further support for the proposed mechanism of action of AN3661 in *S. neurona*.

## AN3661 has a minimal impact on host cell transcription dynamics

A remarkable feature of AN3661 is its selectivity as far as cellular toxicity is concerned. Previously, it was shown that AN3661 had little effect on the growth of cells or animals that serve as hosts for *P. falciparum*, *T. gondii*, and *C. parvum* [7–9]. The results presented in this study show that the drug does not affect the growth of the BT cells used for propagation of *S. neurona*. This reinforces the theme that AN3661 has high specificity for its target in the parasites, and does not affect the host (mammalian) ortholog, CPSF73.

This specificity implies that the drug has no impact on mRNA polyadenylation in host cells or animals. However, the compound JTE-607 (mentioned above) is an inhibitor of CPSF73 function that selectively inhibits the growth of acute myeloid leukemia and Ewing's sarcoma cell lines [24,25], and has been shown to prolong life in a mouse leukemia model [29]. This compound causes wide-ranging changes in poly(A) site usage and transcriptional readthrough in mammalian cells [25]. That a selective compound can alter mRNA polyadenylation raises the possibility that AN3661 may alter 3' end processing, but in ways that have minimal or no impact on host cell physiology. It is thus important that AN3661 had no discernible effect on mRNA polyadenylation and transcription dynamics in BT host cells. In particular, there was no discernible impact of the drug on poly(A) site usage in BT cells (S7 File). Moreover, there were no indications of increased usage of distal poly(A) sites in AN3661-treated BT cells (Fig 8), nor was there evidence for the usage of novel sites downstream of annotated transcription units (S7 File). These parameters (altered poly(A) site choice, increased distal poly(A) site usage, and transcription downstream of annotated genes) are hallmarks of the inhibition of CPSF73 in mammals and yeast [24,25,27,28]. That AN3661 does not alter these features of gene expression in BT cells is strong evidence that the compound in fact does not interfere with CPSF73 functioning in these cells.

The results presented in this report strengthen the case for AN3661 (and related compounds that also target CPSF73 in parasites) as therapeutic agents. However, enthusiasm for such a use is tempered by the scope of mutations in the parasite CPSF73 that can reduce sensitivity to the drug. In *S. neurona*, single mutations near the CPSF73 active site were sufficient to provide resistance to the compound (Fig 2). Resistance also could be achieved by single base changes in *P. falciparum* [7] and *T. gondii* [8]. The reported frequency of spontaneous resistance clones *in vitro* in *P. falciparum* (with resistant clones arising readily in initial populations of as few as $10^6$ cells) [7] is especially notable, as this frequency indicates that resistance would likely arise quickly during a treatment regimen. However, when used in concert with drugs that target different processes, compounds that selectively target CPSF73 in apicomplexans may find use. The absence of effects of the drug on host cells makes such uses attractive, as AN3661 would likely not add any side effects when used in a combination therapy.

## Conclusions

The results reported in this study show that the benzoxaborole AN3661 inhibits the growth of *S. neurona*, an apicomplexan parasite of horses and marine mammals. This inhibition can be attributed to the targeting by the drug of CPSF73. AN3661 had a minimal impact on the bovine host cells used to propagate the parasites; importantly, there was no discernible effect of the drug on poly(A) site choice or overall gene expression in the host cells. Collectively, these results reveal AN3661 to be an exceedingly selective inhibitor of apicomplexan mRNA polyadenylation.

## Supporting information

**S1 Fig. Changes in transcriptional dynamics in *S. neurona* treated with AN3661.** Browser tracks showing examples of changes in poly(A) site usage in AN3661-treated cells. The order for each representation is (top to bottom): chromosomal location (in bp), gene annotation, coding region (CDS) annotation, reads from un-treated cells (Sn3 mapping), and reads from AN3661-treated cells (A90-1 mapping). Reads colored green are oriented in the sense (5'->3' left to right) direction, and reads colored green are oriented in the antisense direction. Tracks were created using CLC Genomics Workbench.
(PDF)

**S1 Table. Primers used in this study.**
(XLSX)

**S1 File. Reads and mapping statistics.**
(XLSX)

**S2 File. Analysis pipeline for poly(A) site determinations.**
(DOCX)

**S3 File. Gene expression analysis in AN3661-treated *S. neurona*.** *S. neurona*-infected BT cells were treated with 90 nM AN3661, RNA isolated, and PATSeq libraries constructed and sequenced. PATSeq reads were used to assess gene expression in drug-treated *S. neurona* cells.
(XLSX)

**S4 File. Poly(A) site analysis in AN3661-treated *S. neurona*.** *S. neurona*-infected BT cells were treated with 90 nM AN3661, RNA isolated, and PATSeq libraries constructed and sequenced. The data were used to assess poly(A) site choice in the parasite in the two conditions.
(XLSX)

**S5 File. Gene expression in control and AN3661-treated BT cells.** RNA was isolated from control and AN3661-treated BT cells and PATSeq libraries constructed and sequenced. PAT-Seq reads were used to assess host gene expression.
(XLSX)

**S6 File. Host gene expression in uninfected and *S. neurona*- infected BT cells.** RNA was isolated from control and *S. neurona*-infected BT cells and PATSeq libraries constructed and sequenced. PATSeq reads were used to assess host gene expression.
(XLSX)

**S7 File. Poly(A) site analysis in AN3661-treated BT host cells.** RNA was isolated from control and AN3661-treated BT cells and PATSeq libraries constructed and sequenced. The data were used to assess poly(A) site choice in the host in the two conditions.
(XLSX)

## Acknowledgments

The authors acknowledge the excellent technical assistance of Carol Von Lanken.

## Author Contributions

**Conceptualization:** Arthur G. Hunt.

**Data curation:** Arthur G. Hunt, Daniel K. Howe, Michelle Yeargan.

**Formal analysis:** Arthur G. Hunt, Daniel K. Howe, Ashley Brown, Michelle Yeargan.

**Funding acquisition:** Arthur G. Hunt, Daniel K. Howe.

**Investigation:** Arthur G. Hunt, Daniel K. Howe, Ashley Brown, Michelle Yeargan.

**Methodology:** Arthur G. Hunt, Daniel K. Howe.

**Project administration:** Arthur G. Hunt, Daniel K. Howe.

**Resources:** Arthur G. Hunt, Daniel K. Howe.

**Software:** Arthur G. Hunt.

**Supervision:** Arthur G. Hunt, Daniel K. Howe, Michelle Yeargan.

**Validation:** Arthur G. Hunt.

**Visualization:** Arthur G. Hunt, Daniel K. Howe.

**Writing – original draft:** Arthur G. Hunt.

**Writing – review & editing:** Arthur G. Hunt, Daniel K. Howe.

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
