## [Decision Letter · Decision Letter 0]

24 Jun 2021

PONE-D-21-14533

Transcriptional dynamics in the protozoan parasite Sarcocystis neurona and mammalian host cells after treatment with a specific inhibitor of apicomplexan mRNA polyadenylation

PLOS ONE

Dear Dr. Hunt,

Thank you for submitting your manuscript to PLOS ONE. After careful consideration, we feel that it has merit but does not fully meet PLOS ONE’s publication criteria as it currently stands. Therefore, we invite you to submit a revised version of the manuscript that addresses the points raised during the review process.

Three expert reviewers have now commented on the manuscript. All found that the manuscript reported data that were worthy of publication. The referees reported issues with the description of data, in particular the treatment of the drug response data, and the description of replicates in these experiments, which should be addressed before publication

We look forward to receiving your revised manuscript.

Kind regards,

Stuart Alexander Ralph

Academic Editor

PLOS ONE

Journal Requirements:

2.Thank you for stating the following in the Acknowledgments Section of your manuscript: 

"Acknowledgements and funding

The authors acknowledge the excellent technical assistance of Carol Von Lanken."

"This research was supported by the USDA National Institute of Food and Agriculture (AGH - Hatch project accession #1020849 and DKH – Hatch project accession 1012262; https://nifa.usda.gov/) and the Amerman Family Equine Research Fund (DKH). The sponsors played no role in the study design, data collection and analysis, decision to publish, or preparation of the manuscript."

Additional Editor Comments (if provided):

Three expert reviewers have now commented on the manuscript. All found that the manuscript reported data that were worthy of publication. The referees reported issues with the description of data, in particular the treatment of the drug response data, and the description of replicates in these experiments, which should be addressed before publication

Reviewers' comments:

Reviewer's Responses to Questions

**Comments to the Author**

1. Is the manuscript technically sound, and do the data support the conclusions?

Reviewer #1: Yes

Reviewer #2: Partly

Reviewer #3: Yes

2. Has the statistical analysis been performed appropriately and rigorously? 

Reviewer #1: No

Reviewer #2: No

Reviewer #3: Yes

3. Have the authors made all data underlying the findings in their manuscript fully available?

Reviewer #1: Yes

Reviewer #2: Yes

Reviewer #3: Yes

4. Is the manuscript presented in an intelligible fashion and written in standard English?

Reviewer #1: Yes

Reviewer #2: Yes

Reviewer #3: Yes

5. Review Comments to the Author

Reviewer #1: The study by Hunt et al. shows the effects of treating Sarcocystis neurona and Neospora caninum with a known inhibitor of the apicomplexan polyadenylation complex. The study is technically sound and underlying sequencing data have been deposited to a public repository. Results of bioinformatics analyses are mostly included in well-documented Supplementary Tables. Before publication, the authors should appropriately analyse the IC50 data and include replicate number. The differential gene expression analysis (as presented in Figure 3B) should also be explained in the Materials and Methods - it was not clear from the methods in Supplementary File S2 how this was performed after read mapping.

I would also suggest moving some of Fig S1 to the main Figures, to complement visualization of the data represented in Figure 5.

Major comments

Fig1A and 1B: The IC50 curves have inconsistent curved lines between data points. The authors should fit an IC50 regression and calculate the IC50 from that.

Minor comments

Lines 101 and 103: Should consistently use g instead of RPM (typo RMP on line 103)

Line 108: Specify that the data are available at the NCBI Sequencing Read Archive under the Bioproject accession PRJNA713353

Line 116: The Supplemental File 2 docx has the heading “Supplemental File 3”

Line 133: Were biological replicates performed for the growth assays? This should be specified (eg n = 1, or n = 3) in the Figure 1 legend. If n >1 error bars should be added or individual datapoints plotted.

Line 143: convention to denote mutation as (for example) Y668N, but using residue numbering for the S. neurona CPSF73.

Line 150: There are some residues in black that are not identical between all 5 sequences, eg residue 519. Change the Figure 2 legend to reflect this.

Figure 7: The tick labels are too blurry to read

Line 518: Italicise species names

Line 539: remove COI disclosure

S2 File: To enable reproducibility, add the code for “tagtrim” to the S2 File or a public repository such as GitHub.

Reviewer #2: This manuscript describes results of S. neurona and N. caninum treated with AN3661. Similar to T. gondii and P. falciparum, AN3661-resistant S. neurona had mutations in the cpsf gene. The study examines the effect of AN3661 on poly(A) sites and transcriptional changes in S. neurona. There are a few suggestions to improve the manuscript, below:

Major points:

1. Figure 1: How many times was this experiment done? There are no error bars here.

2. Fig 1A: % inhibition is calculated compared to control wells containing no drug, which I assume would be 0% inhibition. Why does the curve in panel A (S. neurona) not reach 0% inhibition?

3. Fig 1B: There are no data points around the 50% inhibition range. The data points around the 50% inhibition mark is at 5% and then it jumps to ~98% inhibition. This does not allow for accurate IC50 determination. It looks like there are only 5 concentrations examined. This data would be improved if you added more concentrations, especially around the 50% range.

4. For growth inhibition, parasites were only exposed for 2 h before washing out, and 90 nM AN3661 inhibited 100% when examined 4 days later, but measurements were taken daily? This is confusing.

5. It is unclear how AN3661-R S. neurona parasites were obtained. They were selected with 90 nM AN36661 – from Fig 1 this inhibits 100%? How long was 90nM AN3661 put on AN3661? Not described in materials and methods section.

6. Growth inhibition studies parasites were exposed to 90nM AN3661 for 2h and 4 days later there was no growth. But for poly(A) site profiling parasites were exposed to 90 nM AN3661 for 24hrs. Parasites are then harvested and examined for poly(A) site profiling. Why this concentration and this time frame? It seems that the parasites would be “in the process of dying” – i.e. if you let it go for a couple more days (without drug) the parasites would die (as you show for growth inhibition).

7. How many biological replicates were performed for poly(A) site profiling?

8. Fig 3 legend: how long were S. neurona and B. taurus treated for with 90 nM AN3661? Why was this drug concentration chosen – this inhibits Sn at 100% (Fig 1)

9. Fig 5: what do the error bars represent? Not described in the legend

10. Fig 6: why is LDH positive control twice as much as “max LDH release from lysed cells”? Why are the bar graphs of different width?

11. Fig 8: what do the error bars represent? Not described in the legend

12. Whole genome seqencing was not performed on AN3661-resistant parasites, but rather the researchers amplified the cpsf gene and then sequenced the gene. Therefore would be good to mention in discussion that there could be other genetic mutations underlying AN3661-R.

Minor points:

1. Lines 34-35: this sentence is confusing. If vaccines are ineffective against coccidian parasite how is this an option to reduce coccidiosis?

2. Line 40: EPM not defined

3. Line 103: RPM (typo)

4. The figure legends are peppered in the body of the manuscript.

5. After the first mention of Sarcocystis neurona, future mentions should be S. neurona

6. Line 162: should be bovine turbinate (BT) not bovine (BT) turbinate

7. Line 192: FDR is 0.05 while Line 271 refers to false discovery rate adjusted p-value (i.e. q-value) less than 0.05. BT analysis refers to analysis done on Sn. Was the same analyses performed? Please clarify.

8. Would be good to define “Adjusted p-value” vs :p-value”

Reviewer #3: The manuscript is exceedingly well-written and well-organized with sufficient detail to support scientific claims.

I have only a few comments/questions/suggested edits:

1. The information provided in the introduction could be better supported with citations.

2. Line 52 - affective should be effective

3. Sequencing of the SnCPSF73 gene of the AN3661-resistant clones was performed. Did the authors consider sequencing the full genome of the clones to ensure no other mutant genes were contributing to the phenotype?

4. Line 294 - Typo "BY" host cells, should be "BT"

5. Line 351 - typo "t="

6. Figure 2 - What does the blue star indicate?

7. Figure 3B. x/y axis text not legible, poor resolution.

8. figure 8. (Line 437) Should this be BT cells instead of BY?

9. Figure 8. - Does the "I" in BTI stand for anything? Inhibitor? Because there aren't any parasites in this population of cells consider changing for clarity (e.g., BT-A90).

6. PLOS authors have the option to publish the peer review history of their article (what does this mean?). If published, this will include your full peer review and any attached files.

Reviewer #1: No

Reviewer #2: **Yes: **Caroline Ng

Reviewer #3: No

---

## [Author Response · Author response to Decision Letter 0]

8 Aug 2021

To the Editor,

The following summarizes changes and other discussion regarding the review of manuscript PONE-D-21-14533, “Transcriptional dynamics in the protozoan parasite Sarcocystis neurona and mammalian host cells after treatment with a specific inhibitor of apicomplexan mRNA polyadenylation”, that we have submitted for publication in PLOS ONE. This document is organized using the reviewing outline and format. 

Reviewers' comments:

Reviewer's Responses to Questions

Comments to the Author

1. Is the manuscript technically sound, and do the data support the conclusions?

Reviewer #1: Yes

Reviewer #2: Partly

As indicated in the detailed response, we have made changes to satisfy the concerns regarding statistical analyses and presentation of some data items.

Reviewer #3: Yes

2. Has the statistical analysis been performed appropriately and rigorously? 

Reviewer #1: No

As indicated in the detailed response, we have made changes to satisfy the concerns regarding statistical analyses and presentation of some data items.

Reviewer #2: No

As indicated in the detailed response, we have made changes to satisfy the concerns regarding statistical analyses and presentation of some data items.

Reviewer #3: Yes

3. Have the authors made all data underlying the findings in their manuscript fully available?

Reviewer #1: Yes

Reviewer #2: Yes

Reviewer #3: Yes

4. Is the manuscript presented in an intelligible fashion and written in standard English?

Reviewer #1: Yes

Reviewer #2: Yes

Reviewer #3: Yes

5. Review Comments to the Author

Reviewer #1: The study by Hunt et al. shows the effects of treating Sarcocystis neurona and Neospora caninum with a known inhibitor of the apicomplexan polyadenylation complex. The study is technically sound and underlying sequencing data have been deposited to a public repository. Results of bioinformatics analyses are mostly included in well-documented Supplementary Tables. Before publication, the authors should appropriately analyse the IC50 data and include replicate number. The differential gene expression analysis (as presented in Figure 3B) should also be explained in the Materials and Methods - it was not clear from the methods in Supplementary File S2 how this was performed after read mapping.

We have added the requested information to the Supplementary File S2, as an additional section (Section III). This addition should satisfy the request. Additional text was not added to the Materials and Methods, since this would be redundant. (File S2 has been further edited to fix some other issues and hopefully make it easier to follow.)

The assessment of growth inhibition, including replicate numbers, has been clarified in the Materials and Methods, Results, and the Figure legend, and IC50 values for S. neurona have now been more appropriately determined by regression analysis. Since it was a minor component of the study and the data were not strong, the Neospora growth assay and IC50 determination have been removed from the manuscript. 

I would also suggest moving some of Fig S1 to the main Figures, to complement visualization of the data represented in Figure 5.

After trying to add some browser tracks to Fig. 5, we find that the resulting revised figure is very difficult to read, because of the different formats and styles for the charts and browser tracks. Thus, we have decided to leave Fig. 5 and Fig. S1 unchanged.

Major comments

Fig1A and 1B: The IC50 curves have inconsistent curved lines between data points. The authors should fit an IC50 regression and calculate the IC50 from that.

Our presentation of the growth assay results (Fig. 1) has been revised, and the IC50 for S. neurona has been calculated by regression analysis. We have decided that the N. caninum growth assays were not pursued adequately. Since these data were a minor component of the study, we have elected to remove the old Figure 1B. The new Figure 1B shows results of a S. neurona growth assay using a more narrow range of AN3661 concentrations to better estimate the IC50.

Minor comments

Lines 101 and 103: Should consistently use g instead of RPM (typo RMP on line 103)

RPM has been replaced with x g.

Line 108: Specify that the data are available at the NCBI Sequencing Read Archive under the Bioproject accession PRJNA713353

This change has been made as requested (lines 119-120).

Line 116: The Supplemental File 2 docx has the heading “Supplemental File 3”

This error has been corrected.

Line 133: Were biological replicates performed for the growth assays? This should be specified (eg n = 1, or n = 3) in the Figure 1 legend. If n >1 error bars should be added or individual datapoints plotted.

The number of assays and replicates has been clarified in the figure legend (lines 154-155) and the Materials and Methods (lines 74-79), and the figure has been revised as recommended. 

Line 143: convention to denote mutation as (for example) Y668N, but using residue numbering for the S. neurona CPSF73.

Figure 2 has been revised to account for this, and to add residue numbers appropriate for each sequence that was analyzed.

Line 150: There are some residues in black that are not identical between all 5 sequences, eg residue 519. Change the Figure 2 legend to reflect this.

The legend to Fig. 2 has been revised to better explain the residue shading that was used to depict the alignment.

Figure 7: The tick labels are too blurry to read

The font size for the tick legends have been increased to allow for easier reading.

Line 518: Italicise species names

This correction has been made.

Line 539: remove COI disclosure

This correction has been made.

S2 File: To enable reproducibility, add the code for “tagtrim” to the S2 File or a public repository such as GitHub.

The “tagtrim” program has been deposited on Github and links provided.

Reviewer #2: This manuscript describes results of S. neurona and N. caninum treated with AN3661. Similar to T. gondii and P. falciparum, AN3661-resistant S. neurona had mutations in the cpsf gene. The study examines the effect of AN3661 on poly(A) sites and transcriptional changes in S. neurona. There are a few suggestions to improve the manuscript, below:

Major points:

1. Figure 1: How many times was this experiment done? There are no error bars here.

The number of assays and replicates has been clarified in the text (lines 140-150), figure legend (lines 154-155) and the Materials and Methods (lines 74-79), and the figure has been revised as recommended. 

2. Fig 1A: % inhibition is calculated compared to control wells containing no drug, which I assume would be 0% inhibition. Why does the curve in panel A (S. neurona) not reach 0% inhibition?

Although these data have been re-plotted and the figure revised, the principle is the same. The reviewer is correct that the % inhibition (now % growth) in the treatment wells is relative to the no-drug control wells. Consequently, the control wells are not plotted in the graph; only the treatment data are plotted. Since some inhibition was observed even at the lowest concentration, the curve does not reach 0% inhibition (now 100% growth).

3. Fig 1B: There are no data points around the 50% inhibition range. The data points around the 50% inhibition mark is at 5% and then it jumps to ~98% inhibition. This does not allow for accurate IC50 determination. It looks like there are only 5 concentrations examined. This data would be improved if you added more concentrations, especially around the 50% range.

The reviewer is correct that the growth assay data for Neospora (Fig.1B) was not sufficient. Since Sarcocystis is the primary focus of the study, we have removed the Neospora results. A new Figure 1B is now presented, which shows the growth curve for S. neurona over a more narrow range of drug concentrations.

4. For growth inhibition, parasites were only exposed for 2 h before washing out, and 90 nM AN3661 inhibited 100% when examined 4 days later, but measurements were taken daily? This is confusing.

We apologize for the lack of clarity. The growth assays were conducted with the AN3661 drug present for the duration of the 4-day growth period, not just during the initial 2-hour incubation when parasites were allowed to invade the host cells. This has been clarified in the Materials and Methods (lines 74-87).

Although the plates were monitored daily, only the data from day 4 are presented in Figure 1 and used for the IC50 calculations. This has been clarified in the Materials and Methods (lines 74-87).

5. It is unclear how AN3661-R S. neurona parasites were obtained. They were selected with 90 nM AN36661 – from Fig 1 this inhibits 100%? How long was 90nM AN3661 put on AN3661? Not described in materials and methods section.

The mutagenized population was grown in the presence of AN3661 for approximately 5 weeks, followed by continued selection during the single-cell cloning. This has been clarified in the Materials and Methods (lines 90-96). As mentioned by the reviewer, 90nM was chosen because our growth assays (e.g., Fig. 1) showed that this fully inhibited parasite growth (now mentioned in the Results section, lines 164-167).

6. Growth inhibition studies parasites were exposed to 90nM AN3661 for 2h and 4 days later there was no growth. But for poly(A) site profiling parasites were exposed to 90 nM AN3661 for 24hrs. Parasites are then harvested and examined for poly(A) site profiling. Why this concentration and this time frame? It seems that the parasites would be “in the process of dying” – i.e. if you let it go for a couple more days (without drug) the parasites would die (as you show for growth inhibition).

As clarified above for item #4, the AN3661 drug was present for the entire 4-day growth assay, not just during the initial 2-hour incubation. It is believed that AN3661 is not a parasiticidal drug, but rather just inhibits parasite growth via disruption of normal RNA polyadenylation (the intent of the transcriptome experiment). As we mention above for item #5, 90 nM was chosen because it fully inhibited parasite growth and we wanted to ensure large differences in poly(A) profiling between the treatment sample and the non-treated control sample.

7. How many biological replicates were performed for poly(A) site profiling?

The Materials and Methods (lines 102-109) have been edited to make it clear that three biological replicates were prepared for each condition in the poly(A) site profiling.

8. Fig 3 legend: how long were S. neurona and B. taurus treated for with 90 nM AN3661? Why was this drug concentration chosen – this inhibits Sn at 100% (Fig 1)

As indicated in the Materials and Methods, the AN3661 treatments were done for 24 hr. 90 nM was chosen for the AN3661 concentration for the poly(A) site profiling so as to assure that the two populations (treated and control) were distinct. Use of a concentration nearer the LD50 could have resulted in a mixed population of cells exhibiting different extents of inhibition and transcriptome alterations.

9. Fig 5: what do the error bars represent? Not described in the legend

The error bars represent standard deviations calculated from values in each of the replicates. The figure legend has been edited to make this clear.

10. Fig 6: why is LDH positive control twice as much as “max LDH release from lysed cells”? Why are the bar graphs of different width?

The LDH positive control is an enzyme control provided by the manufacturer to ensure that the assay has been conducted properly. The maximum release sample represents the same number of BT cells as those that have been treated and, therefore, shows the level of LDH release that would be seen if a drug was highly cytotoxic. We neglected to include a description of the LDH release assay in the Materials and Methods section. This has been added (lines 128-135). We have also revised the results and Fig. 6 legend to better explain the results of this experiment.

11. Fig 8: what do the error bars represent? Not described in the legend

The error bars represent standard deviations calculated from values in each of the replicates. The figure legend has been edited to make this clear.

12. Whole genome sequencing was not performed on AN3661-resistant parasites, but rather the researchers amplified the cpsf gene and then sequenced the gene. Therefore would be good to mention in discussion that there could be other genetic mutations underlying AN3661-R.

This point has been raised in the revision (lines 172-175).

Minor points:

1. Lines 34-35: this sentence is confusing. If vaccines are ineffective against coccidian parasite how is this an option to reduce coccidiosis?

This sentence has been revised to specify that vaccination is effective against poultry coccidiosis caused by Eimeria, but has not been effective against other species of coccidia.

2. Line 40: EPM not defined

This has been changed to Equine Protozoal Myeloencephalitis.

3. Line 103: RPM (typo)

This error has been corrected.

4. The figure legends are peppered in the body of the manuscript.

This is the format required for submissions to PLoS ONE.

5. After the first mention of Sarcocystis neurona, future mentions should be S. neurona

This correction has been made.

6. Line 162: should be bovine turbinate (BT) not bovine (BT) turbinate

This correction has been made.

7. Line 192: FDR is 0.05 while Line 271 refers to false discovery rate adjusted p-value (i.e. q-value) less than 0.05. BT analysis refers to analysis done on Sn. Was the same analyses performed? Please clarify.

This statement has been corrected, as suggested. The parameter used in these analyses was indeed the q-value. 

8. Would be good to define “Adjusted p-value” vs :p-value”

Assuming that this refers to the description of the poly(A) site analyses (lines 237-239 and 303-306), the terms “FDR-adjusted p-value” and “raw p-value” have been inserted where needed.

Reviewer #3: The manuscript is exceedingly well-written and well-organized with sufficient detail to support scientific claims.

I have only a few comments/questions/suggested edits:

1. The information provided in the introduction could be better supported with citations.

An additional reference that succinctly reviews the field had been added. 

2. Line 52 - affective should be effective

This correction has been made.

3. Sequencing of the SnCPSF73 gene of the AN3661-resistant clones was performed. Did the authors consider sequencing the full genome of the clones to ensure no other mutant genes were contributing to the phenotype?

We have not done whole-genome sequencing, largely because of cost considerations. Were this the first study of AN3661-resistant apicomplexans, we would concur that whole-genome sequencing and a more exhaustive ruling out of other changes would be called for. However, it is well-established that the changes we report are responsible for AN3661 resistance in other apicomplexans; the consistency we see in multiple mutant isolates suffices to confirm the hypothesis, in our opinion.

4. Line 294 - Typo "BY" host cells, should be "BT"

This error has been corrected.

5. Line 351 - typo "t="

This error has been corrected.

6. Figure 2 - What does the blue star indicate?

These were inadvertent additions to the figure and do not belong. They are carryovers from a different version of the figure, intended to highlight features not discussed in this study.

7. Figure 3B. x/y axis text not legible, poor resolution.

The figure has been modified to improve the legibility and resolution.

8. figure 8. (Line 437) Should this be BT cells instead of BY?

This error has been corrected.

9. Figure 8. - Does the "I" in BTI stand for anything? Inhibitor? Because there aren't any parasites in this population of cells consider changing for clarity (e.g., BT-A90).

We believe the term “BTI” is adequately defined in the legend, so this has not been changed.

6. PLOS authors have the option to publish the peer review history of their article (what does this mean?). If published, this will include your full peer review and any attached files.

Do you want your identity to be public for this peer review? For information about this choice, including consent withdrawal, please see our Privacy Policy.

Reviewer #1: No

Reviewer #2: Yes: Caroline Ng

Reviewer #3: No

---

## [Decision Letter · Decision Letter 1]

31 Aug 2021

PONE-D-21-14533R1

Transcriptional dynamics in the protozoan parasite Sarcocystis neurona and mammalian host cells after treatment with a specific inhibitor of apicomplexan mRNA polyadenylation

PLOS ONE

Dear Dr. Hunt,

Thank you for submitting your manuscript to PLOS ONE. After careful consideration, we feel that it has merit but does not fully meet PLOS ONE’s publication criteria as it currently stands. Therefore, we invite you to submit a revised version of the manuscript that addresses the points raised during the review process.

The referees agree that the revisions made have substantially addressed their concerns, but in the new material there are several areas where minor additional details should be provided for clarity and completeness of the manuscript, as well as some minor typographical and other areas throughout the manuscript that should be addressed.

We look forward to receiving your revised manuscript.

Kind regards,

Stuart Alexander Ralph

Academic Editor

PLOS ONE

Journal Requirements:

Additional Editor Comments (if provided):

The referees agree that the revisions made have substantially addressed their concerns, but in the new material there are several areas where minor additional details should be provided for clarity and completeness of the manuscript, as well as some minor typographical and other areas throughout the manuscript that should be addressed.

Reviewers' comments:

Reviewer's Responses to Questions

**Comments to the Author**

1. If the authors have adequately addressed your comments raised in a previous round of review and you feel that this manuscript is now acceptable for publication, you may indicate that here to bypass the “Comments to the Author” section, enter your conflict of interest statement in the “Confidential to Editor” section, and submit your "Accept" recommendation.

Reviewer #1: (No Response)

Reviewer #2: (No Response)

Reviewer #3: All comments have been addressed

2. Is the manuscript technically sound, and do the data support the conclusions?

Reviewer #1: Yes

Reviewer #2: Yes

Reviewer #3: Yes

3. Has the statistical analysis been performed appropriately and rigorously? 

Reviewer #1: Yes

Reviewer #2: Yes

Reviewer #3: Yes

4. Have the authors made all data underlying the findings in their manuscript fully available?

Reviewer #1: Yes

Reviewer #2: Yes

Reviewer #3: Yes

5. Is the manuscript presented in an intelligible fashion and written in standard English?

Reviewer #1: Yes

Reviewer #2: Yes

Reviewer #3: Yes

6. Review Comments to the Author

Reviewer #1: The authors have satisfactorily addressed my comments. Prior to publication, the authors should ensure that the "tagtrim" GitHub repository is set to public as the address https://github.com/ArthurGHunt/tagtrim currently leads to a 404 page.

Reviewer #2: The revised manuscript is much improved, especially in regards to statistical analyses, graphical representations, and clarification of methods. Here are some suggestions to further improve the manuscript, detailed below:

Major Comments:

1. Define abbreviations at first use. Some examples (please go through to identify others):

a. CPSF73 in the abstract Line 4

b. BT cells Line 70 Methods

c. FDR Line 240

d. YSH1 Line 381

e. IPA1 Line 383

2. Reference 1 – can’t find this in PubMed. Would be good to include doi for all articles not in PubMed.

3. Statements in the introduction are largely unsupported by the necessary references. Please include a reference for each statement made, e.g. for statements in lines 24-32, lines 38-48, 50-51

4. Methods: Please include more details on the LDH assay – eg what wavelength was this assay read at? What is the composition of the lysis buffer?

5. How many total drug-resistant single-cell clones were obtained? (in reference to lines 167-168)

6. Fig. 4: what does DEGs stand for? What does the blue circle and the yellow circle indicate? Unclear from the figure and legend.

7. “The results (Fig. 4) show that between 24 and 27% of genes 247 whose poly(A) site profiles change also showed significant changes in overall transcript levels.” – I don’t see these percentages in the figure.

8. Fig. 6: What is the measurement that was taken? As written it looks like a measurement was taken at Abs 490 through Abs 680 nm. A quick look into the manufacturer’s protocol suggests that this is not the case. Please correct the figure itself to more accurately reflect what was done, and also describe this in the figure legend.

9. Statement on Lines 418-420 incorrect: studies were performed with various initial populations of P. falciparum; the least amount of parasites was 2 x 107

Minor Comments:

1. Line 70: Space between cloneswere

2. Line 81: 37oC

3. Be consistent: hours denoted as “h” in line 80 but “hrs” on lines 103,104 and elsewhere

4. Fig 1 legend: Line 158: nM; line 160: IC50, and needs units after 14.99.

5. Fig 2: “Hs” is not aligned with the other names. Suggest writing Tg, Sn, Pf, At, Hs for consistency. Information about strains that this sequence comes from should be addressed in the legend. – i.e. please explain TGME49, SN3, Pf3D7

6. Line 193 – (BT) after “bovine turbinate” and not “bovine”

7. Line 340: benzoxaborole (typo)

8. Line 427: no studies on N. caninum anymore in this report

9. Fig 3 legend: S. neurona and B. taurus need to be italicized

Reviewer #3: (No Response)

7. PLOS authors have the option to publish the peer review history of their article (what does this mean?). If published, this will include your full peer review and any attached files.

Reviewer #1: No

Reviewer #2: **Yes: **Caroline Ng

Reviewer #3: No

---

## [Author Response · Author response to Decision Letter 1]

29 Sep 2021

To the Editor,

The following summarizes changes and other discussion regarding the second review of manuscript PONE-D-21-14533, “Transcriptional dynamics in the protozoan parasite Sarcocystis neurona and mammalian host cells after treatment with a specific inhibitor of apicomplexan mRNA polyadenylation”, that we have submitted for publication in PLOS ONE. This document is organized using the reviewing outline and format. In so doing, our changes, responses, and other discussion are set apart after each section of comments and suggestions.

Major Comments:

1. Define abbreviations at first use. Some examples (please go through to identify others):

a. CPSF73 in the abstract Line 4

b. BT cells Line 70 Methods

c. FDR Line 240

d. YSH1 Line 381 – no need to change, since this is a gene name

e. IPA1 Line 383 - no need to change, since this is a gene name

The changes in lines 4, 70, and 240 of the first revision have been made as suggested. USH1 and IPA1 are formal gene designations and not abbreviations. Accordingly, these have not been changed or further defined.

2. Reference 1 – can’t find this in PubMed. Would be good to include doi for all articles not in PubMed.

The doi for this book has been added. We believe the other references are in Pubmed.

3. Statements in the introduction are largely unsupported by the necessary references. Please include a reference for each statement made, e.g. for statements in lines 24-32, lines 38-48, 50-51

We appreciate this remark. However, these statements are all general parasitology textbook information (as could be found in the first reference). We would argue that references to the literature, or other textbooks, are appropriate for this paragraph. Accordingly, we have not added any.

4. Methods: Please include more details on the LDH assay – eg what wavelength was this assay read at? What is the composition of the lysis buffer?

This section of the Methods has been updated as requested. We would note that the lysis buffer was provided with the commercial assay. The composition of the buffer is not provided in the documentation by the manufacturer. 

5. How many total drug-resistant single-cell clones were obtained? (in reference to lines 167-168)

This statement has been updated as requested.

6. Fig. 4: what does DEGs stand for? What does the blue circle and the yellow circle indicate? Unclear from the figure and legend.

The legend to Figure 4 has been edited so as to clarify these questions. In particular, in the legend we now note that the yellow circles (identifiable as “DEG” in the figure) corresponds to differentially-expressed genes.

7. “The results (Fig. 4) show that between 24 and 27% of genes 247 whose poly(A) site profiles change also showed significant changes in overall transcript levels.” 

We have not altered the figure, out of the belief that to add percentages to the figure in addition to or instead of the total gene numbers would be confusing, and would not allow readers to appreciate the scope of the changes we describe. It is relatively simple for a reader to divide the numbers in the overlap by the numbers of total genes in each class, to come to the percentages we mention.

8. Fig. 6: What is the measurement that was taken? As written it looks like a measurement was taken at Abs 490 through Abs 680 nm. A quick look into the manufacturer’s protocol suggests that this is not the case. Please correct the figure itself to more accurately reflect what was done, and also describe this in the figure legend.

We have corrected the Y axis label and defined the terms in the revised figure legend.. 

9. Statement on Lines 418-420 incorrect: studies were performed with various initial populations of P. falciparum; the least amount of parasites was 2 x 107 . 

From Ref 7: “We quantified the ease of in vitro selection of resistance to AN3661 by subjecting 106–108 Dd2 strain parasites to 60 nM (2 × IC90) AN3661. Regrowth was seen in two of three cultures with initial inocula of 106 parasites at days 45 and 56, one of three cultures with initial inocula of 107 parasites at day 23, and three of three cultures with initial inocula of 108 parasites, all on day 19 (Supplementary Table 4). These rates of resistance selection were similar to those observed for atovaquone.”

Minor Comments:

1. Line 70: Space between cloneswere

This correction has been made.

2. Line 81: 37oC

This correction has been made.

3. Be consistent: hours denoted as “h” in line 80 but “hrs” on lines 103,104 and elsewhere

We now use hrs throughout..

4. Fig 1 legend: Line 158: nM; line 160: IC50, and needs units after 14.99.

These corrections have been made.

5. Fig 2: “Hs” is not aligned with the other names. Suggest writing Tg, Sn, Pf, At, Hs for consistency. Information about strains that this sequence comes from should be addressed in the legend. – i.e. please explain TGME49, SN3, Pf3D7

The figure has been slightly modified, and the legend edited, to address this issue. We would note that the gene identifiers for the proteins have been added to the Methods.

6. Line 193 – (BT) after “bovine turbinate” and not “bovine”

This correction has been made.

7. Line 340: benzoxaborole (typo).

This correction has been made.

8. Line 427: no studies on N. caninum anymore in this report

This correction has been made.

9. Fig 3 legend: S. neurona and B. taurus need to be italicized 

This correction has been made.

---

## [Editor Report · Decision Letter 2]

13 Oct 2021

Transcriptional dynamics in the protozoan parasite Sarcocystis neurona and mammalian host cells after treatment with a specific inhibitor of apicomplexan mRNA polyadenylation

PONE-D-21-14533R2

Dear Dr. Hunt,

We’re pleased to inform you that your manuscript has been judged scientifically suitable for publication and will be formally accepted for publication once it meets all outstanding technical requirements.

Kind regards,

Stuart Alexander Ralph

Academic Editor

PLOS ONE
---

## [Editor Report · Acceptance letter]

19 Oct 2021

PONE-D-21-14533R2 

Transcriptional dynamics in the protozoan parasite *Sarcocystis neurona* and mammalian host cells after treatment with a specific inhibitor of apicomplexan mRNA polyadenylation 

Dear Dr. Hunt:

I'm pleased to inform you that your manuscript has been deemed suitable for publication in PLOS ONE. Congratulations! Your manuscript is now with our production department. 

Kind regards, 

on behalf of

Dr. Stuart Alexander Ralph 

Academic Editor

PLOS ONE